# Familial Hemiplegic Migraine with an ATP1A4 Mutation: Clinical Spectrum and Carbamazepine Efficacy

**DOI:** 10.3390/brainsci10060372

**Published:** 2020-06-15

**Authors:** Giangennaro Coppola, Grazia Maria Giovanna Pastorino, Luigi Vetri, Floriana D’Onofrio, Francesca Felicia Operto

**Affiliations:** 1Clinic of Child and Adolescent Neuropsychiatry, Department of Medicine, Surgery and Dentistry, Medical School University of Salerno, 84100 Salerno, Italy; graziapastorino@gmail.com (G.M.G.P.); floriana.donofrio09@gmail.com (F.D.); opertofrancesca@gmail.com (F.F.O.); 2Department of Health Promotion, Mother and Child Care, Internal Medicine and Medical Specialties (PROMISE), University of Palermo, 90133 Palermo, Italy; luigi.vetri@gmail.com

**Keywords:** familial hemiplegic migraine, *ATP1A4* gene, carbamazepine, clinical symptoms

## Abstract

An Italian family with familial hemiplegic migraine (FHM) with the absence of mutations in the known genes associated with this disorder, namely *ATP1A2*, *ATP1A3*, *CACNA1A*, and *SCN1A*, has recently been reported. Soon afterward, whole exome sequencing allowed the identification of the carrier status of a heterozygous *ATP1A4* mutation c.1798 C >T, in four affected members of this family. Here we compare the clinical symptoms of the affected family members with those from the other FHM families linked to mutations in the known genes associated with this disorder. A further two-year follow-up, including clinical response to carbamazepine administered to the proband and the maternal grandmother due to a worsening of the migraine symptoms, is reported. The clinical condition of the proband’s brother, carrying the same mutation and suffering from congenital ventricular and supraventricular extrasystoles, isdiscussed as well.

## 1. Introduction

An Italian family with familial hemiplegic migraine (FHM) [1] with the absence of mutations in the known genes associated with this disorder, namely *ATP1A2*, *ATP1A3*, *CACNA1A*, and *SCN1A*, has recently been reported.

Soon afterward, a whole exome sequencing combined with Sanger sequencing carried out in all the affected family members led to the identification of the carrier status of a previously not reported heterozygous *ATP1A4* mutation [2].

In the present work, the clinical symptoms described in this family are compared with those from the FHM families carrying mutations in the known genes associated with this disorder. A further two-year follow-up, including the clinical response to carbamazepine administered to the proband and the maternal grandmother due to a worsening of the migraine symptoms, has been made. The clinical picture of the proband’s brother, carrying the same mutation and suffering from congenital ventricular and supraventricular extrasystoles, is discussed as well.

## 2. Family Report (See Figure 1, Family Pedigree)

### 2.1. I,2: Female, Aged 78 Years

The patient started to have recurrent moderate to severe migraine attacks lasting about 1–2 h at the age of 23 years. Each episode was preceded by a visual aura, often triggered by stress factors. In the last four months, headache attacks, lasting about 1 to 3 h, increased up to three/day, and were associated with amaurosis. A single daily dose of carbamazepine led to a significant clinical improvement. Headache was also responsive to the early administration of ibuprofen 60 mg. An ambulatory electrocardiogram (ECG) at the age of 70 years disclosed 421 supraventricular ectopic beats (SVEBs).

### 2.2. II,2: Female, 52 Years of Age

Migraine attacks started at six years of age, characterized by a visual aura (light flashing or scotoma) followed by paresthesia and weakness of the right arm and right side of the face and tongue, and then spreading to the right lower limb. The episodes lasted 2 to 24 h and were associated with vomiting, drowsiness, and lethargy. In adolescent age, the patient suffered from extremely severe episodes, recurring threeto fourtimes each year. From the age of 22 years, the patient had rare migraine attacks (about oneper year). After the age of 49 years, symptoms became more severe. In the last two years, the patient reported two to three episodes of migraine with amaurosis responsive to single doses of ibuprofen.

### 2.3. II,4: Female, 48 Years of Age

The patient reported his first migraine attacks at twelve years. They consisted of a visual aura, including flashing lights mainly in the right eye, followed in a few minutes by weakness/paresis of the right arm and lower limb associated with intense and disabling headache. Migraine episodes were generally responsive to the early administration of ibuprofen. Migraine attacks were initially three to four per year, falling to one/year at the age of 41 years and on. In the last year, the patient was still suffering from monthly episodes of hemiplegic migraine.

### 2.4. III,1: Male, 19 Years of Age

At the age of 3 years, the child was prescribed a 24-h holter ECG that reported 741 ventricular ectopic beats (VEBs) and 7136 supraventricular ectopic beats (SVEBs) without structural heart abnormalities. Heart arrhythmia persisted unchanged, not requiring any drug treatment. A heart magnetic resonance imaging (MRI scan with particular regard to the seno-atrial node could not be performed up to now because of the patient’s tachyarrhythmia that proved to be resistant to any medication.

### 2.5. III,2: Male Proband, 16 Years of Age

The first migraine attacks appeared at the age of nineand half years, upon awakening, characterized by pulsating pain in the right side of the frontal region, associated with a blurred vision and paresthesias/weakness on the right body side. The episodes lasted about half an hour. More than twenty episodes occurred in the following 2 months, thus leading the child to stop school lessons. Brain computed tomography CT/MRI scans and electroencephalographic (EEG) recordings, together with lab tests, were normal. In the hypothesis of a familial hemiplegic migraine, genetic workup including array comparative genomic hybridization (array-CGH) and sequencing of the main genes associated with this disorder (*ATP1A2*, *ATP1A3*, *SCN1A*, *CACNA1A*, *PRRT2*, *SLC1A4*, *and SLC4A4*), was carried out with no significant results. 

A whole exome sequencing (WES) performed soon after disclosed a heterozygous *ATP1A4* gene mutation c.1798 C>T.

The patient underwent neuropsychological assessment through standardized tests (WISC-IV, EpiTrack Junior) which showed a normal intellectual profile (110 as total IQ), while a mild deficit of the executive functions (attention, working memory, inhibition, shifting), probably related to the headache recurrence, was found. The emotional/behavioral profile with Child Behavior Check List (CBCL) didnot show pathological scores.

In the last 3–4 months, the patient reported daily episodes of intense dizziness leading him to sit down for a few minutes. Migraine attacks with transient amaurosis and right-side hemiplegia occur about each month. The episodes last a few hours and usually disappear upon falling asleep.

### 2.6. III,3: Male, Age of 12 Years

At this age, the clinical history showed rare episodes of severe, disabling headache without visual aura or other accompanying symptoms, especially when attending school activities. 

## 3. Discussion

In the present study, a new family is described with the aggregation of four relatives affected by hemiplegic migraine, covering three successive generations. 

After excluding the involvement of the known genes associated with FHM, such as *CACNA1A*, *ATP1A2*, *ATP1A3*, and *SCN1A* in the affected patients, a whole exome sequencing was performed that led to the identification of a rare mutation (frequency 0.0004) predicted as deleterious in a “new” gene, namely *ATP1A4.*

The same ATP1A4 mutation was also found in a brother of the index patient who exhibited several ventricular and supraventricular ectopic beats with childhood-onset. No mutation was found in any of the healthy controls.

The diagnosis of FHM in our family appears to be satisfied by the presence of typical symptoms, including at least two migraine attacks preceded by visual symptoms and transient unilateral weakness and paresthesias. Symptoms were almost overlapping in the index patient and in three other first or second-degree relatives, thus suggesting an autosomal dominant transmission. 

Other potential causes of headache, such astransient ischemic attacks and stroke, were ruled out in all our patients, as well as other migraine disorders following the International Classification of Headache Disorders-3 (ICHD-3) criteria [3]. Overall, in this family, the hemiplegic migraine can be considered a form with less severe expressivity, improving with increasing age, when the attacks become rare, and the motor symptoms may be even absent.

With respect to the known *FHM* genes, *CACNA1A* (HFM1) is mostly associated with cerebellar signs, ataxia, dysarthria, and nystagmus and possible neurological damage, such ascerebral atrophy, whereas *ATP1A2* mutations (HFM2) have been found in some patients with severe symptoms including prolonged (up to several weeks), motor deficit and sometimes coma and epilepsy or cognitive deficits [4].

Most of the FHM3 patients, due to *SCN1A* gene mutations, report only visual symptoms, consisting of recurrent reflex daily blindness [5,6].

*PRRT2* mutations reported in families with a spectrum of movement and/or convulsive disorders and sporadic cases with hemiplegic migraine have been ruled out in our family, as well as *ATP1A3 and SLC2A1* gene mutations associated with alternating hemiplegic migraine, which usually has a very early onset with a different clinical presentation [7]. Mutations in *SLC1A3 and SLC4A4* genes observed, respectively, in sporadic cases and in two sisters with hemiplegic migraine in co-morbidity with other disorders, such as seizures, episodic ataxia, renal tubular acidosis, and ocular anomalies [8,9], were also excluded.

*KCNK18* gene mutations reported in another family with multiple cases of migraine with aura [10], were ruled outas well. Of interest is the condition of *ATP1A4* mutation carrier of the family member III, 1, brother of the index patient, who has never had migraine attacks up to the age of 17 years and is suffering from a cardiac arrhythmia diagnosed at the age of 3 years. It consists of several and persistent ventricular and supraventricular ectopic beats, in the absence of any structural heart abnormality. No drug therapy has been given to date.

The condition of the family member III, 1, could be explained as due to an incomplete penetrance of the *ATP1A4* gene, similarly to what happens for *ATP1A2* and *CACNA1A* genes (penetrance 67–87%) or to a random association or to a pleiotropic effect.

Interestingly, *ATP1A4* gene is expressed not only in the testis, in the brain and in the kidney, but also in the myocardium and the cardiovascular system. A potential role in the genesis of this early-onset idiopathic tachyarrhythmia cannot, therefore, be completely excluded. This gene mutation might, therefore, be looked for in patients with such a disorder of unknown origin. Noteworthy, a further 2-year follow-up highlighted a moderate increase in migraine attacks, which significantly improved with a low daily dose of carbamazepine in the proband and his grand-mother.

Carbamazepine was given considering its efficacy in familial and sporadic cases of hemiplegic migraine associated with *PRRT2* gene mutations [11,12]. To our knowledge, no cases of hemiplegic migraine linked to mutations in other known genes have been treated with this drug so far. Although it is quite difficult to draw any hypothesis on the efficacy of carbamazepine, it can be argued that Na, K-ATPase isoforms, including alpha four, are influenced by changes in extracellular Na (+) and that they may play a role in membrane potential [13]. Carbamazepine may, therefore, be considereda drug of choice for the treatment of hemiplegic migraine.

## 4. Conclusions

In the present family, clinical symptoms are suggestive of a familial hemiplegic migraine, though with an overall milder expressiveness compared to other FHM cases associated with mutations on ionic channels or other ATP exchangers. Interestingly, paresthesias and transient weakness interested the right body side in all relatives. 

The association between heart arrhythmia and the *ATP1A4* mutation in the proband’s brother may be considered as a random event or due to an incomplete penetrance of the mutation or to a pleiotropic effect.

This gene should be sequenced in FHM families negative for mutations in the genes associated with this disorder. Finally, sequencing of the *ATP1A4* gene might also be considered in subjects with congenital ventricular and supraventricular extrasystolic beats of unknown origin.

## Figures and Tables

**Figure 1 brainsci-10-00372-f001:**
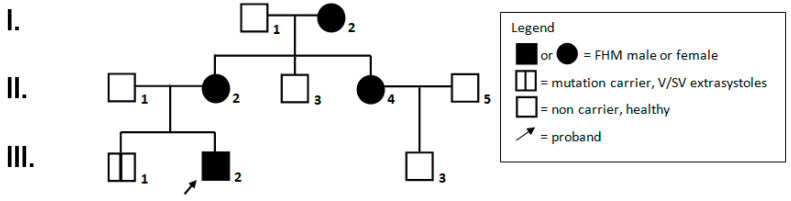
Family pedigree.

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
