# Peer review of "Familial Hemiplegic Migraine with an ATP1A4 Mutation: Clinical Spectrum and Carbamazepine Efficacy"

_brainsci, 2020, doi:10.3390/brainsci10060372_

Round 1
Reviewer 1 Report
The paper is well written. I agree with the type of study design used.
Very useful and appreciated evaluation of clinical symptoms through follow-up.
I would ask to review English language of the paper.
Author Response
Response to Reviewer 1 Comments
Point 1: The authors are grateful to the reviewer 1 who suggested to improve English style of the manuscript-
Response 1: A tracked version of the manuscript was provided showing all the changes made to the text.

Reviewer 2 Report
The manuscript presented details a case report of a family with Familial Hemiplegic Migraine with ATP1A4 mutation.
The data presented in the manuscript is largely identical to that presented in a previous manuscript by the authors (Palumbo et al, 2019, Arch Clin Med Case Rep). The reference to follow-up and Carbamazepine efficacy in the title is only covered briefly in the text, with minimal details. Unfortunately I don't think there is sufficient or significant new data presented that would support this publication.
Author Response
Response to Reviewer 2 Comments
Point 1: The Authors are grateful to the reviewer for his valuable suggestion to comment more extensively on the efficacy of CBZ in two members of this family. In this regard, the following was added in the discussion section:
Response 1: Discussion section (line 166): “Carbamazepine was given considering its efficacy in familial and sporadic cases of hemiplegic migraine associated with PRRT2 gene mutations [11,12]. To our knowledge, no cases of hemiplegic migraine linked to mutations in other known genes have been treated with this drug so far. Although it is quite difficult to draw any hypothesis on the efficacy of carbamazepine, it can be argued that Na, K-ATPase isophorms including alpha 4 are influenced by changes in extracellular Na (+)and that they may play a role in membrane potential [13]. Carbamazepine may therefore be considered a drug of choice for the treatment of hemiplegic migraine.”
The following were added to the text and in References section:
- Dale, R.C., Gardiner, A., Branson, J.A, Houlden H. Benefit of carbamazepine in a patient with hemiplegic migraine associated with PRRT2 mutation. Dev Med Child Neurol. 2014;56(9):910.
- Suzuki-Muromoto, S., Kosaki, R., Kosaki, K., Kubota, M. Familial hemiplegic migraine with a PRRT2 mutation: Phenotypic variations and carbamazepine efficacy. Brain Dev. 2020;42(3):293-297
- Jimenez, , Sánchez, G., Wertheimer, E.V., Blanco, G. Activity of the Na,K-ATPase alpha4 Isoform Is Important for Membrane Potential, Intracellular Ca2+, and pH to Maintain Motility in Rat Spermatozoa. Reproduction 2010;139(5):835-45
Point 2: With regard to the comment on the overlap of this manuscript with the report by Palumbo et .al (2019), it should be considered that while the previous work is largely focused on the genetic finding, this one is essentially dedicated to the comparison (in Discussion section) of the clinical symptoms in this family with what is reported in families with mutations in other known genes linked to this disorder. A two -year follow-up has then been added which highlights, in this family, persistence and in some cases even worsening of symptoms and carbamazepine efficacy in two family members. In Discussion section, the index patient’ brother who resulted as well carrier of the same mutation, has been added; interestingly, this child is suffering from a congenital idiopathic heart arrhythmia, not reported in the first work.

Reviewer 3 Report
Comments:
Abstract:
Line 14: the abbreviation (FHM) should be added after “familial hemiplegic migraine”.
Introduction:
Line 27: the abbreviation (FHM) should be added after “familial hemiplegic migraine”.
Line 27: “..recently been reported by some of us..” Please, rephrase this sentence.
Although the aim of the article by Coppola et al. is stated clearly, the introduction could be improved.
Family report:
The meaning of the abbreviations BESV (line 46) and EKG (line 65) should be detailed.
Figure 1:
The meaning of the square alone should be provided.
Is it correct to represent the square 3 (3 generation) under circle 4 (generation 2)?
Author Response
Response to Reviewer 3 Comments
Response to Reviewer 3 Comments
Point 1: Abstract:
Line 14: the abbreviation (FHM) should be added after “familial hemiplegic migraine”.
Response 2: Line 14.” Recently, an Italian family with familial hemiplegic migraine (FHM)”
Point 2: Introduction:
Line 27: the abbreviation (FHM) should be added after “familial hemiplegic migraine”.
Response 2: Line 27: An Italian family with familial hemiplegic migraine (FHM)
Point 3: Line 27: “..recently been reported by some of us..” Please, rephrase this sentence.
Although the aim of the article by Coppola et al. is stated clearly, the introduction could be improved.
Response 3: The introduction section has been changed as follows:
“ Hemiplegic migraine, a rare form of migraine with recurrent weakness as a motor aura, can occur as a sporadic or a familial disorder, linked to mutations in at least four different genes involved in ion transportation. Familial hemiplegic migraine has an autosomal dominant mode of inheritance. The clinical pattern of sporadic and familial cases with identified mutations is characterized by phenotypic heterogeneity, varying from pure hemiplegic migraine to severe early-onset forms with transient or permanent neurologic disorders. To date, at least a quarter of affected families and most sporadic cases are negative for mutations in the known genes, suggesting that other genes are still to be identified [1].
An Italian family with familial hemiplegic migraine (FHM) [1]has recently been reported by some of us in which with absence of mutations in the known genes associated with this disorder, namely ATP1A2, ATP1A3, CACNA1A and SN1A, were excluded has recently been reported.
A subsequent Soon afterwards, a whole exome sequencing combined with Sanger sequencing carried out in all the affected family members led to the identification of the carrier status of a nunpreviously described not reported heterozigous ATP1A4 mutation [2].
In Tthe present work makes a comparison of the clinical symptoms described in this family are compared with those from the other FHM families linked to other gene carrying mutations in the known genes associated with this disorder. A further two-year follow-up,has also been made including the clinical response to carbamazepine administered to the proband and the maternal grandmother due to a worsening of the migraine symptoms, has been made. The clinical picture of the proband’s brother, carrying the same mutation and suffering from congenital ventricular and supraventricular extrasystoles, is also as well discussed.”
Point 4: Family report:
The meaning of the abbreviations BESV (line 46) and EKG (line 65) should be detailed.
Response 4: Line 46 An ambulatory electrocardiogram (ECG) at the age of 70 years disclosed 421
supraventricular ectopic beats (SVEBs).
Line 65 At the age of 3 years, the child underwent a 24-h holter ECG that reported 741 ventricular ectopic beats (VEBs) and 7136 supraventricular ectopic beats (SVEBs)
Point 5 : Figure 1:
The meaning of the square alone should be provided.
Is it correct to represent the square 3 (3 generation) under circle 4 (generation 2)?
Response 5: Fig. 1 has been changed (see the Tracked version of the manuscript)
Point 6: English style needs to be improved
Response 6: please read the tracked version of the manuscript with all the changes made to the text
Point 1: Abstract:
Line 14: the abbreviation (FHM) should be added after “familial hemiplegic migraine”.
Response 2: Line 14.” Recently, an Italian family with familial hemiplegic migraine (FHM)”
Point 2: Introduction:
Line 27: the abbreviation (FHM) should be added after “familial hemiplegic migraine”.
Response 2: Line 27: An Italian family with familial hemiplegic migraine (FHM)
Point 3: Line 27: “..recently been reported by some of us..” Please, rephrase this sentence.
Although the aim of the article by Coppola et al. is stated clearly, the introduction could be improved.
Response 3: The introduction section has been changed as follows:
“ Hemiplegic migraine, a rare form of migraine with recurrent weakness as a motor aura, can occur as a sporadic or a familial disorder, linked to mutations in at least four different genes involved in ion transportation. Familial hemiplegic migraine has an autosomal dominant mode of inheritance. The clinical pattern of sporadic and familial cases with identified mutations is characterized by phenotypic heterogeneity, varying from pure hemiplegic migraine to severe early-onset forms with transient or permanent neurologic disorders. To date, at least a quarter of affected families and most sporadic cases are negative for mutations in the known genes, suggesting that other genes are still to be identified [1].
An Italian family with familial hemiplegic migraine (FHM) [1]has recently been reported by some of us in which with absence of mutations in the known genes associated with this disorder, namely ATP1A2, ATP1A3, CACNA1A and SN1A, were excluded has recently been reported.
A subsequent Soon afterwards, a whole exome sequencing combined with Sanger sequencing carried out in all the affected family members led to the identification of the carrier status of a nunpreviously described not reported heterozigous ATP1A4 mutation [2].
In Tthe present work makes a comparison of the clinical symptoms described in this family are compared with those from the other FHM families linked to other gene carrying mutations in the known genes associated with this disorder. A further two-year follow-up,has also been made including the clinical response to carbamazepine administered to the proband and the maternal grandmother due to a worsening of the migraine symptoms, has been made. The clinical picture of the proband’s brother, carrying the same mutation and suffering from congenital ventricular and supraventricular extrasystoles, is also as well discussed.”
Point 4: Family report:
The meaning of the abbreviations BESV (line 46) and EKG (line 65) should be detailed.
Response 4: Line 46 An ambulatory electrocardiogram (ECG) at the age of 70 years disclosed 421
supraventricular ectopic beats (SVEBs).
Line 65 At the age of 3 years, the child underwent a 24-h holter ECG that reported 741 ventricular ectopic beats (VEBs) and 7136 supraventricular ectopic beats (SVEBs)
Point 5 : Figure 1:
The meaning of the square alone should be provided.
Is it correct to represent the square 3 (3 generation) under circle 4 (generation 2)?
Response 5: Fig. 1 has been changed (see the Tracked version of the manuscript)
Point 6: English style needs to be improved
Response 6: please read the tracked version of the manuscript with all the changes made to the text

Round 2
Reviewer 2 Report
The authors have made some improvements to the text as per reviewers suggestions. However, I feel there is still no fundamental difference between this manuscript and the authors previously published 2019 case study. The addition of carbamazepine information is limited, and the authors indeed state in their conclusions that "it is quite difficult to draw any hypothesis on the efficacy of carbamazepine...".
Author Response
Point 1: The authors have made some improvements to the text as per reviewers suggestions. However, I feel there is still no fundamental difference between this manuscript and the authors previously published 2019 case study. The addition of carbamazepine information is limited, and the authors indeed state in their conclusions that "it is quite difficult to draw any hypothesis on the efficacy of carbamazepine...".
Response 1: I am sorry to disagree with the reviewer 2, but in our opinion the effectiveness of carbamazepine is itself a very interesting and useful data for the clinician who must face this type of disorder. Some members of this family have suffered for years from several, sometimes severe, hemiplegic migraine attacks, with a symptomatic and not always helpful approach. It is interesting for its quick efficacy at low doses and for being effective in cases of hemiplegic migraine from mutation other than the already known PRRT2. It would be interesting to evaluate CBZ also in HM cases from mutations in other known genes as ATP1A2 ATP1A3, CACNA1A; but this data is not reported to our knowledge. Any proof of efficacy of CBZ in migraine is somewhat significant when considering that “CBZ is among AEDs which have insufficient evidence for use in migraine prevention” (Simy K Parikh 1, Stephen D Silberstein 2.Current Status of Antiepileptic Drugs as Preventive Migraine Therapy. 2019 Mar 18;21(4):16.).
It is confirmed that it is difficult to make definitive hypotheses on the effectiveness of CBZ, but this is always and in all cases (not only in headache, but in focal epilepsy too). It should not be forgotten that CBZ as an antiepileptic drug was born by serendipity, in subjects with trigeminal neuralgia and epilepsy. We have, however, prudently added in the text some hypotheses on the potential mechanism of action of CBZ with regard to the ATP1A4 mutation in this family.

This manuscript is a resubmission of an earlier submission. The following is a list of the peer review reports and author responses from that submission.